## [Decision Letter · Decision Letter 0]

27 Oct 2025

PGENETICS-D-25-01012

PARP-2 catalytic activity drives replication-ICL repair in an allele-specific manner during germline development

PLOS Genetics

Dear Dr. Kim,

Thank you for submitting your manuscript to PLOS Genetics. After careful consideration, we feel that it has merit but does not fully meet PLOS Genetics's publication criteria as it currently stands. Therefore, we invite you to submit a revised version of the manuscript that addresses the points raised during the review process.

Please submit your revised manuscript as soon as possible, ideally within 4 months. If you will need more time than this to complete your revisions, please reply to this message or contact the journal office at plosgenetics@plos.org. Please include the following items when submitting your revised manuscript:

We look forward to receiving your revised manuscript.

Kind regards,

Stefan Taubert, PhD

Academic Editor

PLOS Genetics

Marnie Blewitt

Section Editor

PLOS Genetics

Aimée Dudley

Editor-in-Chief

PLOS Genetics

Anne Goriely

Editor-in-Chief

PLOS Genetics

**Additional Editor Comments:**

As you will see, the reviewers agreed that the work was of potential interest to the readers of PLoS Genetics. However, revisions are required to clarify the nature and specifics of reagents used and experiments performed; some experiments need to be expanded in scope, including the addition of important controls for comparison purposes; some experiments need quantification for better evaluation; some data may need re-interpretation, including depending on new requested experiments (e.g., levels vs. localization); and in some cases clarification should be added in the data presentation. Please see the reviewer's comments for details.

**Journal Requirements:**

At this stage, the following Authors/Authors require contributions: Xiaojing Ren, Anna Hu, Zifei Liu, Semin Kim, and Hyun-Min Kim. Please ensure that the full contributions of each author are acknowledged in the "Add/Edit/Remove Authors" section of our submission form.

The list of CRediT author contributions may be found here: https://journals.plos.org/plosgenetics/s/authorship#loc-author-contributions

4) Please amend your detailed Financial Disclosure statement. This is published with the article. It must therefore be completed in full sentences and contain the exact wording you wish to be published.

2) If any authors received a salary from any of your funders, please state which authors and which funders..

**Reviewers' comments:**

Reviewer's Responses to Questions

**Comments to the Authors:**

Reviewer #1: The manuscript by Ren et al., characterizes the role of PARP-2 and other PARylation genes in the C. elegans germ line. Using genetic, cytological, and molecular analyses the authors provide evidence for both a catalytic and structural role of PARP-2 in normal cells and in response to DNA damage. Overall, this is an interesting study with implications for the role of PARylation and the use of PARP inhibitors The following should be addressed:

One issue with following the narrative is the inconsistent use of the different PARylation mutants throughout the study: Figure 1 and 2 presents data on parp-2. Figure 3 examines parp-1, parp-2, and parg-1. Figure 4 and 5 examines parp-1, papr-2 and parg-1 for some of the assays, but not all. Figure 6 and 7 presents data on parp-2. I have more specific comments below but it does make it confusing to follow. I don’t think everything has to be done with all the mutants but in some cases it would be beneficial to include additional data on the different mutants.

Figure 1: Please provide quantification of the westerns. It would be very useful to include asterisks or brackets to direct the reader to the specific part of the immunoblot referred to in the text. The authors indicate that there is a complete loss of PARylation based on Figure 1E but the data is not convincing (at least to me). Given the partially redundant roles of parp-1 and parp-2 it seems that the single parp-1 and double parp-1; parp-2 mutant should be examined in this assay.

Figure 2: Please indicate F = protein and G = mRNA directly on graphs. On line 186, the authors state “Relocalization of PARP-2 upon UV and HU” and in both the results and the discussion indicate that PARP-2 becomes enriched in the nucleolus. I do not see this in the data: while it is convincing that PARP-2 is enriched in the nucleolus in late meiotic prophase, this does not appear to be the case upon UV/HU treatment. If I have this wrong, then you will have to show more convincing images. Otherwise, I think the data is consistent with elevated levels upon treatment. (not relocalization).

Figure 3: The authors show that parg-1 produces a high incidence of males – please note that this has been published in Janisiw et al., 2020 and this should be stated and referenced.

Figure 4: It is unclear why the authors only examined the putative null of parp-2, parp-1 and parg-1 with UVC and HU and then examined both parp-2 alleles but not parp-1 or parg-1 with nitrogen mustard and cisplatin. It seems to me that all the assays should be performed with all the mutants.

Figure 5: Please show both parp-2 mutants, parp-1, and parg-1 for both pCHK1 and ATL-1.

Figure 6: Please show RAD-51 immunolabeling for both parp-2(ok344) and parp-2(E509K) as was done for FCD-2. It says there is elevated RAD-51 in both mutants in the figure legend but as far as I can tell only one is examined and labeled (parp-2).

Minor issues:

Line 304: hating should be hatching

Line 393: flies should be worms

Line 462: italicize rad-51 and pif-1.

Reviewer #2: In their article, Ren et al. investigated the role of PARP-2 in specific DNA repair pathways using C. elegans as a model system. They found that PARP-2 plays a particularly important role in the response to replicative DNA damage, as exemplified by treatment with hydroxyurea, and proposed possible mechanistic explanations. Overall, the manuscript is clear and well-structured, and the assays employed are consistent with the standards of the field. Since the relative roles of different PARPs in maintaining cellular homeostasis are not yet fully understood, this article would make an important contribution to the respective field. Before I can recommend this manuscript for publication, several points require clarification.

Major points.

1. Figures 1c and 1d examine the consequences of the PARP-2 deletion shown in Figure 1a on gene and protein expression. Since most of the gene is deleted in the ok344 allele (panel a), it is unclear why residual expression is still observed at the mRNA level (Figure 1c) and, in particular, at the protein level (Figure 1d). One would expect at least a strongly truncated protein to appear on the western blot; however, the data instead show a full-length protein, albeit at reduced levels. Could the authors clarify this discrepancy?

2. The authors use the data in Figure 6 to propose that PARP-2 cooperates with FCD-2 in mediating ICL repair. However, the evidence presented is not conclusive. To demonstrate specificity, the authors should perform co-stainings with mediators of at least one additional DNA repair pathway, taking for instance the examples shown in Figure 7E. Without such controls, it cannot be excluded that PARP-2 interacts more generally with DNA repair pathways, rather than specifically with ICL repair.

3. The authors suggest that a catalytically inactive variant of PARP-2 is more effective in inducing the DNA damage response, citing enhanced expression of a broad range of DNA repair pathways in the E509K mutant as key evidence (Figure 7e). However, Figure 6c shows reduced association between E509K variant and FCD-2 compared to the full-length PARP-2. Taken together, the data in Figures 7e and 6c would instead suggest that catalytically inactive PARP-2 is impaired in inducing a functional DNA damage response, which may be compensated by broad upregulation of repair pathways. The authors should clarify this apparent discrepancy or reconsider their proposed model.

4. Figure 7c and the accompanying text suggest that PARP-2 may be involved in ribosome biogenesis; however, the presented data (partial nucleolar localization and modest changes in the expression of certain related genes) are insufficient to establish this conclusively. To provide definitive evidence, functional assays are required, such as measurements of rRNA levels and ribosome content, or assessments of overall translation and de novo translation (e.g., puromycin-labelling assays).

Reviewer #3: The authors present a manuscript on C. elegans parp-2. The authors demonstrate association of epitope-tagged PARP-2 with the nucleus at some stages of germ cell and embryo development. In addition, the authors report a modest increase in embryo lethality for both parp-2 and parp-1 mutants, but there is a pronounced ~60% embryo lethality for the E509K catalytic inactivation transgene of parp-2. The authors also observed that all E509K animals that hatch arrest as larvae, but that larval arrest is observed for 13% of E509K / + animals. The authors observe modest reductions in fertility for parp-1, parp-2 and parg-1 mutants, with modest reductions in germline size and pachytene for parp-2 and parg-1. Modest sensitivity to UV light is reported, strong sensitivity to HU for parp-2 and moderate sensitivity to nitrogen mustard and cisplatin for both parp-2 deletion and parp-2 catalytic dead. The authors nicely show that CHK-1 and ATL-1 respond to DNA damage in parp-1, parp-2 and parg-1 mutant germ cells in the absence of DNA damaging agents. The authors do a good job of showing that multiple concentrations of human PARP inhibitors affect CHK-1 activation, suggesting DNA damage upon loss of PARP activity. RAD-51 foci are observed in mitotic germ cells upon parp-2 loss. CHIP seq and genomics analysis nicely shows PARP-2 enrichment at transcription start sites, possibly at GAGA motifs. The authors note that DDR gene expression is affected in parp mutants.

Overall, this paper could be improved if the authors clearly explain how the mutations that they study were characterized and if the E509K phenotype is zygotic or maternal. The CHIP seq of PARP-2 in wild type may provide information about PARP function as a transcriptional regulator, which might be consistent with the mutant phenotypes. Localization to GATA sites at promoters might explain the strong E509K defects, but the authors need to more clearly compare the promoters that PARP-2 binds to with genes that are misregulated in order to ascribe a PARP-2 function in activation or silencing of gene expression. This might offer a new model for mammalian PARP, which is among the most abundant proteins in the nucleus, and if conserved might be important when considering PARPi use for cancer treatment.

Comments

1. The authors state that ‘Western blot analysis showed that while wild type animals expressed a strong band at the predicted size . . . ‘ Please state in the text what antibody is being used.

2. ‘interestingly, transgenic animals expressing the E509K mutant-lack of PARylation activity, showed a complete loss of PARylation.’ It is difficult to understand this experiment from the text. I first thought ‘why would PARylation be absent from animals expressing a KD version of parp-2.’ The authors need to state in the text that they created an ok344 strain that expressed E509K parp-2, but that this did not have parylation activity. However, it is also not clear what kind of transgene the authors used. Please state what promoter was used, and if the transgene is single copy or multicopy in the Results, so the ready can assess the type of experiment being conducted. Also, how do the authors know that the E509K mutation is causing lack of PARylation? They need to test wild type transgene to know that their transgene construct can rescue PARylation in order to conclude that the E509K mutation causes lack of parylation.

3. The authors state ‘PARP-2 is essential for restraining nucleolar expansion’. Expansion suggests an increase in nucleolar size. However, the authors report an increase in NOP-1 fluorescence that could be independent of nucleolar size. Perhaps the authors should measure nucleolar size in oocytes to ask if parp-2 affects this variable.

4. The NOP-1 decline in wild type animals and in parp-2 mutants in -1 oocytes is consistent with disappearance of the nucleolus in -1 oocytes.

5. ‘In contrast, the FLAG::HA::PARP-2 transgenic reporter showed no significant difference in embryo lethality, suggesting preserved functionality’. Is this a transgene or an endogenous parp-2 tag. If it is a transgene, is parp-2 wild type or is the transgene in a parp-2 ok344 mutant background, allowing the authors to conclude ‘preserved functionality’. Please clarify these points in the Results text.

6. The authors suggest that E509K catalytic inactivation mutation of parp-2 causes very high levels of embryo lethality because it is a ‘toxic gain of function’. However, e509K / + embryos survived at wild type levels ‘demonstrating clear haplosufficiency’. I am not sure what the authors mean by haplosufficiency, perhaps that one wild type parp-2 allele rescues the E509K defect. This indicates that E509K is a recessive mutation.

7. The authors also observed that all E509K animals that hatch arrest as larvae, but that larval arrest is observed for 13% of E509K / + animals. The authors need to state here how they know that the animals they are scoring are E509K / +. Are they using a GFP-marked balancer chromosome? If so, what is the frequency of larval arrest for this balancer chromosome when in trans to wild type.

8. If one observes a phenotype like a Him phenotype for a single parg-1 allele that is induced by a mutagen-induced mutation like EMS or UV-TMP, this needs to be confirmed with a second allele or by RNAi or by rescue of the mutant phenotype. This is why clearly stating that the FLAG::parp-2 transgene is rescuing the ok344 mutation for embryo lethal phenotype is important to state in the text of the Results.

9. In the Methods, there is no indication of how the deletion mutation strains were processed. Were all deletions verified as being homozygous using one primer pair that detects the deletion and a second primer pair that only detects wild type DNA? The reason for this genotyping is that some strains sent from a stock center can be heterozygous for a deletion mutation. How many times were the deletions outcrossed versus wild type prior to conducting the reported experiments. As discussed above, mutagens that induce ~16 homozygous loss of function mutations per genome were used to create these deletions. Any of these mutations could cause the reported phenotypes associated with deletion mutations. For parp-2, the authors may be able to use a second mutant allele or rescue by transgene. The parp-2 catalytic dead strain is useful to confirm the effects of nitrogen mustard and cisplatin sensitivity of parp-2 deletion, but the catalytic dead mutant was not used to confirm the effects on HU. However, the parp-2 catalytic dead has strong phenotypes of its own. It might be helpful to confirm the parp-2 deletion phenotype of strong HU sensitivity with a null parp-2 allele, which can be easily created by CRISPR/Cas9 using the STOP-IN cassette described by Sternberg.

10. The authors report E905K parp-2 mutants display reduced germ cell nuclei in response to HU, suggesting the S phase check point is intact. How are the authors measuring germ cell nuclei if all KD mutant animals arrest as larvae?

11. The authors report that human parp inhibitors lead to reduced parp-2 mRNA expression. Why would compounds that inhibit PARP catalytic activity result in reductions in RNA expression from parp genes?

12. The authors observe reduced PARP-2 signal intensity in pachytene but not mitotic regions of germ cells in response to PARP inhibitors. They claim that this could be due to differential tissue penetration, but this is unlikely to be the case because the germ line is a syncytium that has a common cytoplasm. Also, what do the authors mean by reduced signal intensity? Are the PARP inhibitors in human known to reduce the levels of PARP by 50%? Or do these inhibitors simply inhibit PARP catalysis independent of protein levels? Please update the text to state the premise of the experiment. Another possibility is that PARP levels in nuclei were reduced, but it is not clear that the authors are measuring nuclear PARP-2 but rather overall PARP-2 fluorescence.

13. Increased apoptosis is observed in parp mutant germ lines. This is nice, but please explain how the apoptosis was measured in the text (currently not in text or figure legend).

14. Much higher levels of FCD-2 foci are observed in parp mutants. Perhaps confirm with a parp inhibitor?

15. The authors map the locations of PARP-2 in the genome (in the absence of DNA damages) and suggest a role in GAGA promoters. Then, the authors perform RNA seq in parp-2 mutants and note changes to DDR gene expression. Please add a Venn diagram to indicate the overlap between promoters that PARP-2 binds and genes that PARP-2 regulates, for genes that are up or down-regulated. Does this suggest that PARP-2 is a repressor or activator of transcription. For panel 7C show Venn diagrams with overlap for parp-2 deletion and parp KD mutant gene expression changes. It is possible that the expression differences are due to the embryo lethal and fully penetrant larval arrest phenotypes of parp-2 KD - explain in the Results text how the authors prepared the RNA - was it from the same developmental stages? Was RNA prepared from the KD mutant before larval arrest or while larvae were growing? Were dead embryos included in the KD RNA preps?

16. Several GAGA TFs have been associated with aging, including ELT-6. Perhaps show Venn diagram with binding sites for these TFs and those of PARP-2.

17. When introducing the CHIP-seq and genomics experiments in the Results, please indicate if mammalian PARP is known to bind specific segments of the genome or to act as a transcription factor. This will help the reader understand if the authors genomics experiments are creating a new model for how PARP might function.

18. line 563 ‘E509K mutant results in phenotypes that are comparable to or even more sever than those observed in the null allele, including reduced brood size, high embryo lethality and complete larval arrest.’ If the E509K mutant results in complete larval arrest, how does it result in reduced brood size? Perhaps the embryo arrest and complete larval arrest phenotypes occur for F3 progeny of E509K -/- F2 homozygous mutant adults? If so, this phenotype is maternal effect embryo lethality and larval arrest.

19. The authors should indicate in the Results what larval stage the E509K mutants arrest at and what the larvae look like. Are they skinny dauer larvae? Do they arrest as tiny L1 larvae? Are the larvae normal in appearance or are they stiff and unmoving? How do the larval arrest phenotypes of the E509K mutant compare to those of the other parp deletion mutants?

20. In the discussion of E509K mutant phenotypes, the authors should comment on if the PARP inhibitors cause similar larval arrest and embryo lethal phenotypes. This might be expected if the E509K mutation interferes with PARP catalysis similar to a PARP inhibitor.

21. Might the parg-1 deletion suppress the phenotypes of the E509K mutant? If so, the authors could test this. It is probably also worth asking if PARP inhibitors suppress the E509K phenotypes, in case the E509K causes hyperactivation of rather than inactivation of PARP.

22. It would be nice to confirm the effectiveness of the PARP inhibitors using HU assays, where parp-2 mutants show high levels of embryo lethality. Similarly, it would be nice to ask if the parg-1 mutant phenotype of X chromosome missegregation is due to excess PARylation by asking if this phenotype is suppressed by the PARP inhibitors.

23. Methods for creation of the tagged parp-2 need to be added. Was the endogenous locus targeted by CRISPR/Cas9? What guide RNA was used? What DNA oligonucleotide was used to create the tag? Was the sequence of the insertion validated?

**Have all data underlying the figures and results presented in the manuscript been provided?**

Reviewer #1: Yes

Reviewer #2: Yes

Reviewer #3: **No:** The methods used for genetic analysis need to be clearly stated.

PLOS authors have the option to publish the peer review history of their article (what does this mean? ). If published, this will include your full peer review and any attached files.

**Do you want your identity to be public for this peer review?** For information about this choice, including consent withdrawal, please see our Privacy Policy .

Reviewer #1: No

Reviewer #2: No

Reviewer #3: No

**Figure resubmission:**
---

## [Decision Letter · Decision Letter 1]

20 Jan 2026

PGENETICS-D-25-01012R1

PARP-2 catalytic activity drives replication-ICL repair in an allele-specific manner during germline development

PLOS Genetics

Dear Dr. Kim,

Thank you for submitting your manuscript to PLOS Genetics. After careful consideration, we feel that it has merit but does not fully meet PLOS Genetics's publication criteria as it currently stands. Therefore, we invite you to submit a revised version of the manuscript that addresses the points raised during the review process.

Please submit your revised manuscript within by Feb 19 2026 11:59PM. If you will need more time than this to complete your revisions, please reply to this message or contact the journal office at plosgenetics@plos.org. Please include the following items when submitting your revised manuscript:

We look forward to receiving your revised manuscript.

Kind regards,

Stefan Taubert, PhD

Academic Editor

PLOS Genetics

Marnie Blewitt

Section Editor

PLOS Genetics

Aimée Dudley

Editor-in-Chief

PLOS Genetics

Anne Goriely

Editor-in-Chief

PLOS Genetics

**Additional Editor Comments:**

As you can see the reviewers were pleased with the overall improvements of the manuscript, however, there are a few remaining minor issues that require attention. Please address these as outlined in the reviewer's comments.

**Journal Requirements:**

1) We note that your Data Availability Statement is currently as follows: "All relevant data are within the manuscript and its Supporting Information files.". Please confirm at this time whether or not your submission contains all raw data required to replicate the results of your study. Authors must share the “minimal data set” for their submission. PLOS defines the minimal data set to consist of the data required to replicate all study findings reported in the article, as well as related metadata and methods (https://journals.plos.org/plosone/s/data-availability#loc-minimal-data-set-definition).

**Reviewers' comments:**

Reviewer's Responses to Questions

**Comments to the Authors:**

Reviewer #1: The revised manuscript by Ren et al., characterizes the role of PARP-2 and other PARylation genes in the C. elegans germ line. Using genetic, cytological, and molecular analyses the authors provide evidence for both a catalytic and structural role of PARP-2 in normal cells and in response to DNA damage. The authors have done a good job responding to the previous reviews. However, there a still a few minor points that need to be addressed:

While the authors acknowledge that PARP-2 relocalizes to the nucleolus in untreated worms and have rewritten the results to reflect that, both the abstract and author summary still indicate that relocalization occurs upon replication stress. See lines 14-15 and 33, please change accordingly.

Figure 1: The analysis of PARP by immunoblot has been improved; however, the result section could be clearer (lines 140-141). I recommend acknowledging that there is still signal in the absence of parp-2 (and parp-2; parp-1(RNAi)) and this could be a consequence of the antibody recognizing non-specific epitopes and/or that there is residual PARP in the absence of these genes. Another possibility is that the RNAi was not completely effective – did the authors measure RNAi effectiveness? I couldn’t find anything in the materials and methods.

Figure 2: The finding that HU exposure increases the levels of PARP-2 in both the PMT and pachytene (lines 220-222) needs to be discussed as there is not bulk DNA replication at pachytene.

Lines 414-416 are not complete sentences.

Reviewer #2: The authors have addressed all of my previous comments by providing clarifications and rephrasing parts of the manuscript to avoid overstatements and overinterpretation of the experimental data. I am satisfied with the changes made and have no further questions.

Reviewer #3: The authors have done a reasonable job of responding to reviewers’ comments and with improving the structure and content of their manuscript. Importantly, they more clearly describe the genetics and Crispr modifications that they use to study parp mutants, which is essential to understanding what they have done and what their results mean. The Methods can still be improved, to help the reader understand the genetic background of each deletion strain studied, by explicitly stating that the deletion strains were outcrossed or not outcrossed. It might also be helpful if the L1 larval arrest and embryo lethality phenotypes could be reported for parp inhibitors. Overall, this manuscript ties together a role for PARP-2 in responding to endogenous forms of DNA damage, and to DNA replication blocks.

Comments:

1. ‘Homozygous E509k/E509K worms derived from E509K / mIn1 heterozygous parents developed normally and reached adulthood, but their progeny exhibited high embryonic lethality and larval arrest.’

The above could be clarified by stating: ‘Homozygous F2 E509k/E509K worms derived from E509K / mIn1 heterozygous parents developed normally and reached adulthood, but their F3 progeny exhibited high embryonic lethality and larval arrest.

2. ‘All measurements were performed using this second-generation progeny.’

I am not sure about the above comments. My understanding is that fertility was measured for F2 E509K -/- homozygous F2 adults, and then that some data also concern the embryo lethality and larval arrest phenotypes of F3 progeny.

Perhaps instead ‘Fertility and IF measurements were performed using fertile second-generation F2 progeny, whereas developmental abnormalities of embryo lethality and larval arrest were measured for F3 progeny.’

3. ‘Heterozygous E509K / + embryos however survived at near wildtype levels, indicating that E509K behaves as a recessive, catalytically inactive mutation.’ Did the authors measure embryo lethality of E509K / + embryos or of E509K / E509K F2 homozygous embryos and their E509K / + heterozygous siblings? If so, this indicates that the role of PARP-2 catalysis in embryonic development is maternally rescued, but that complete loss of PARP-2 catalysis in F3 embryo results in embryo lethality and larval arrest.

4. ‘HIM’ phenotype in lines 290 and 294 should be ‘Him’. C. elegans phenotypes are capitalized for the first letter, whereas protein names are all capital letters, i.e., HIM-8 protein versus Him phenotype of him-8 mutants.

5. ‘The parp-1 mutant did not exhibit changes in germline length’. Please add this data to the manuscript.

6. line 361 ‘E509K mutants also showed significantly reduced survival compared to wildtype’. Please state the generation tested. Are the authors treating F2 E509K / E509K homozygous mutants and then scoring the survival of F3 progeny? If so, the F3 progeny display substantial embryo lethality, so please explain to the reader how the reduced embryo survival was scored, where normalized levels of embryo lethality may have been substantially higher for E509K F3 progeny.

7. The Methods are improved with verification of the parp-2 deletion by PCR. However, PCR verification of the other deletion mutants is missing from the Methods. Also, it is important to indicate for each deletion strain how many times, if any, the strain was outcrossed prior to analysis by the investigators. This is important because mutagens used to create the parp-2 and other deletions create many mutations. Each deletion strain is estimated to possess 16 loss of function mutations. Therefore, any phenotype reported by the authors could be due to a parp or parg deletion, or to other mutations.

Based on the above logic, the authors need to clearly state in the methods how many times a deletion mutation was outcrossed. For example, ‘We verified that each deletion mutation was homozygous by PCR, and we did not outcross our deletion mutant strains prior to performing the experiments discussed here.

8. The authors suggest that parp inhibitors mimic effects of parp-2 deletion mutation or perhaps more importantly the E509K catalytic inactive mutant. However, the parp inhibitor phenotypes reported in figure 5 do not include embryo lethality or larval arrest, which are pronounced phenotypes of the E590K mutant. Since the parp-2 E509K mutant is maternally rescued, the authors should report effects applying parp inhibitors for two generations. There may be a reason why the authors do not report effects of parp inhibitors on embryo lethality or larval arrest, possibly because both parp-1 and parp-2 are inhibited, which might lead to complete embryonic lethality in the next generation.

9. Supplementary figure S4 does not show the shared impact (or lack thereof) of parp-2 deletion and parp-2 E509K. A Venn diagram should be added to Figure 7C, showing the number of up and down-regulated genes for both parp-2 deletion and E509K strains, and the overlap. Figure S5 shows a number of genes that are upregulated for these strains, but it would be helpful to know how many genes display shared changes to expression for parp-2 deletion and parp2 E509K.

**Have all data underlying the figures and results presented in the manuscript been provided?**

Reviewer #1: Yes

Reviewer #2: Yes

Reviewer #3: Yes

PLOS authors have the option to publish the peer review history of their article (what does this mean? ). If published, this will include your full peer review and any attached files.

**Do you want your identity to be public for this peer review?** For information about this choice, including consent withdrawal, please see our Privacy Policy .

Reviewer #1: No

Reviewer #2: **Yes:** Maria Ermolaeva

Reviewer #3: No

**Figure resubmission:**
---

## [Editor Report · Decision Letter 2]

19 Feb 2026

Dear Dr Kim,

We are pleased to inform you that your manuscript entitled "PARP-2 catalytic activity drives replication-ICL repair in an allele-specific manner during germline development" has been editorially accepted for publication in PLOS Genetics. Congratulations!

Yours sincerely,

Stefan Taubert, PhD

Academic Editor

PLOS Genetics

Marnie Blewitt

Section Editor

PLOS Genetics

Aimée Dudley

Editor-in-Chief

PLOS Genetics

Anne Goriely

Editor-in-Chief

PLOS Genetics

BlueSky: @plos.bsky.social

Comments from the reviewers (if applicable):

**Data Deposition**

http://datadryad.org/submit?journalID=pgenetics&manu=PGENETICS-D-25-01012R2

**Press Queries**

---

## [Editor Report · Acceptance letter]

PGENETICS-D-25-01012R2

PARP-2 catalytic activity drives replication-ICL repair in an allele-specific manner during germline development

Dear Dr Kim,

We are pleased to inform you that your manuscript entitled "PARP-2 catalytic activity drives replication-ICL repair in an allele-specific manner during germline development" has been formally accepted for publication in PLOS Genetics! Your manuscript is now with our production department and you will be notified of the publication date in due course.

With kind regards,

Anita Estes

PLOS Genetics

On behalf of:
